# The Effect of Increasing Irrigation Rates on the Carbon Isotope Discrimination of Apple Leaves

Jan Haberle [1,*], Ivana Raimanová [1], Pavel Svoboda [1], Michal Moulik [1], Martin Mészáros [2] and Gabriela Kurešová [1]

1   Department of Sustainable Arable Land Management and Cropping Systems, Crop Research Institute, 16106 Prague, Czech Republic; raimanova@vurv.cz (I.R.); svoboda@vurv.cz (P.S.); michal.moulik@vurv.cz (M.M.); kuresova@vurv.cz (G.K.)
2   Technology Department, Research and Breeding Fruit Institute, 50801 Holovousy, Czech Republic; martin.meszaros@vsuo.cz
*   Correspondence: haberle@vurv.cz; Tel.: +420-702087697

**Abstract:** $^{13}C$ discrimination ($\Delta^{13}C$) has been used in research as an indicator of water availability in crops; however, few data are available concerning fruit trees. The aim of this study was to examine the effect of irrigation on the $\Delta^{13}C$ values of apple leaves. We assumed that $\Delta^{13}C$ would increase with irrigation intensity. The $\Delta^{13}C$ of apple trees (*Malus domestica*) cv. 'Red Jonaprince' was determined in the years 2019–2022. Leaf samples were collected in spring, in June, at the beginning of the irrigation campaign, and in autumn, in September, following the harvest. The irrigation doses were applied to replenish the water consumption, 0% (ET0), 50% (ET50), 75% (ET75), and 100% (ET100), of the calculated evapotranspiration (ET) levels. In November, the leaves collected from different positions on the shoots were sampled, assuming the $\Delta^{13}C$ signature would reflect the changes occurring in the water supply during their growth. The irrigation rates had a significant effect on the $\Delta^{13}C$ of the leaves when the data for the spring and summer months were pooled. On average, $\Delta^{13}C$ increased from 20.77‰ and 20.73‰ for ET0 and ET50, respectively, to 20.80‰ and 20.95‰ for ET75 and ET100, respectively. When the data obtained for the spring and summer months were analysed separately, the effect of irrigation was weak ($p < 0.043$). The $\Delta^{13}C$ value was always higher for treatment ET100 than treatment ET0, for individual experimental years and terms; however, the differences were minor and mostly insignificant. The leaf position had a strong significant effect on $\Delta^{13}C$; the values gradually decreased from the leaves growing from two-years-old branches (22.50‰) to the youngest leaves growing at the top of the current year's shoots (21.07‰). This order was similar for all the experimental years. The results of the experiment suggest that $^{13}C$ discrimination in apples is affected not only by water availability during growth, but also by the use of C absorbed in previous years.

**Keywords:** $\Delta^{13}C$; $\delta^{15}N$; AWC; evapotranspiration; leaf position





## 1. Introduction

The supply of available water is one of the main limiting factors of yield in non-irrigated, rain-fed agriculture on much of the world's agricultural land. Fluctuations in precipitation and increasing evapotranspiration levels due to higher temperatures result in the frequent occurrence of periods of water shortage during the growth of crops with impacts on the yields and yield stability. The demand for irrigation water is increasing; however, water supplies provided for irrigation purposes are likely to decrease, even in regions such as central Europe, with mild, temperate climates [1–4]. The high intensity of cultivation and climate changes increase the demands for the supplemental irrigation of fruit trees, hops, and vines, which, in the past, were not grown under irrigation. Irrigation doses and timing for fruit trees in farm praxis are based on the tree's requirements for water, obtained from the empirical data for given soil climate and production conditions or

calculated as water loss by the evapotranspiration process. Less frequently, the need for supplemental irrigation is derived from soil moisture sensors or plant data [5–8]. Frequent droughts and the shortage of water resources, especially in the 21st century, have resulted in the pressure to conserve water used for irrigation purposes. More data on this subject are needed, to allow for a comprehensive assessment of the irrigation effects on fruit trees.

An opportunity to indirectly assess water availability to and management by plants provides the $^{13}C$ discrimination ($\delta^{13}C$, $\Delta^{13}C$) of plants. Plants reduce water losses through transpiration by closing their stomata. This modifies the concentrations of $CO_2$ present inside their leaves and leads to changes in the $^{13}C$ and $^{12}C$ isotopic ratio [9]. The $\delta^{13}C$ value is calculated from the $^{12}C$ and $^{13}C$ contents; for the description of the changes occurring in the $^{13}C$ discrimination, $\Delta^{13}C$ is computed from $\delta^{13}C$ using the isotopic composition of the atmosphere [10,11]. The enrichment of the leaf tissue with $^{13}C$ indicates moisture deficits. The $\Delta^{13}C$ value has been used in recent decades as an indicator of water stress and as a possible selection trait for selecting tolerant or adaptable crop genotypes, including fruit trees [6,12–14]. The natural discrimination of $^{13}C$ integrates water availability effects on a plant in the long term, which has its advantages and disadvantages in comparison with the methods that determine the actual status of plants. According to Glenn [15], the $^{13}C$ response is not linked to short-term environmental conditions but, rather, to the tree's accumulated response to them via stomatal regulation. A higher $\Delta^{13}C$ value (corresponding to a better water supply) suggests greater photosynthetic activity as a precondition for attaining a higher yield. As a result, a correlation between the yield and $\Delta^{13}C$ value was observed in crops, especially when plants grown under strong stress and optimal conditions or under irrigation were compared [12,16].

In comparison to field crops, fewer $\Delta^{13}C$ data have been published in the literature for fruit trees, wines, or hops in relation to water supply and irrigation activities [6,13,15,17]. For example, Brillante et al. [18] proposed $\Delta^{13}C$ as an alternative to traditional measurements of water status to capture the spatial variability of the physiological traits at the vineyard scale. Several authors have shown that $^{13}C$ discrimination is related to water use efficiency (WUE) in herbaceous plants and also in fruit trees [6,15,17,19]. In the case of apple trees, the assessment of WUE, a key element of irrigation management, is complicated by the commonly practiced fruit thinning during growth [20–22].

The $^{13}C$ discrimination cannot be used for operational irrigation management, but can provide useful additional information on the effect of water supply over a longer time series, possibly even from the previous growing season. Interpretation of soil moisture or leaf water status monitoring data also requires the integration of instantaneous values, including assigning different weights to these values over the course of the day and over longer periods. The main objective of this study was to examine the effect of increasing irrigation doses on the $\Delta^{13}C$ of apple leaves. We assumed that the $\Delta^{13}C$ signature significantly distinguished between treatments with irrigation from a rain-fed control, apple trees dependent only on precipitation.

We hypothesise that high irrigation rates balance the differences in leaf $\Delta^{13}C$ values during the experimental years (differing in water supply from rainfall), compared to non-irrigated trees. We also verify whether different supplies of irrigation water during the growth stage can be reflected in the $\Delta^{13}C$ signatures of leaves of the different positions on apple shoots.

## 2. Materials and Methods

### 2.1. Site Characteristics

The four-year experiment (2019–2022) was conducted in Holovousy (50.3733847 N, 15.5798914 E), in East Bohemia, the Czech Republic. The experimental apple orchard of the Research and Breeding Institute of Pomology Holovousy is situated 302 m above sea level on Haplic Luvisol soil, and the slope value is 2.09°. The water table is approximately 5 m deep. Soil and agrochemical conditions are presented in Table 1. The soil characteristics

were the averages obtained from soil samples collected in the years 2019 and 2020; the wilting point was calculated from the soil texture data.

**Table 1.** Soil and agrochemical data for the orchard located in Holovousy.

| Soil Layer | Texture | Volume Weight | FWC [(1)] WP [(2)] | AWC [(3)] | Corg Total N | pH (KCl) | Available Nutrients [(4)] P, K, and Mg |
|---|---|---|---|---|---|---|---|
| cm | - | g m$^{-3}$ | % vol. | % vol. | % | - | Mg kg$^{-1}$ |
| 0–30 | Silt loam | 1.45 | 31.9 13.3 | 18.6 | 1.53 0.16 | 6.32 | 124.20 260.80 214.70 |
| 30–60 | Silt loam | 1.42 | 34.1 14.1 | 20.0 | 0.90 0.10 | 6.42 | 11.60 147.50 191.20 |
| 60–90 | Silt loam | 1.45 | 33.0 14.9 | 18.1 | 0.33 0.05 | 6.37 | 1.60 121.80 162.50 |

[(1)] Field water capacity; [(2)] wilting point; [(3)] available water capacity; [(4)] P, K, and Mg, Mehlich III.

The site was located in a temperate-climate region, with a mean annual temperature (1964–2021) of 8.8 °C and rainfall level of 664 mm; the respective mean values of the experimental years (2019–2022) were 9.7 °C and 606 mm (Figure 1). Reference evapotranspiration, ET (Penman–Monteith equation) was, on average, 560 mm, with a maximum month sum calculated from June to August (81–95 mm). The water balance level was calculated as the precipitation minus ET. The negative water balance and deficit levels mostly occurred during summer.

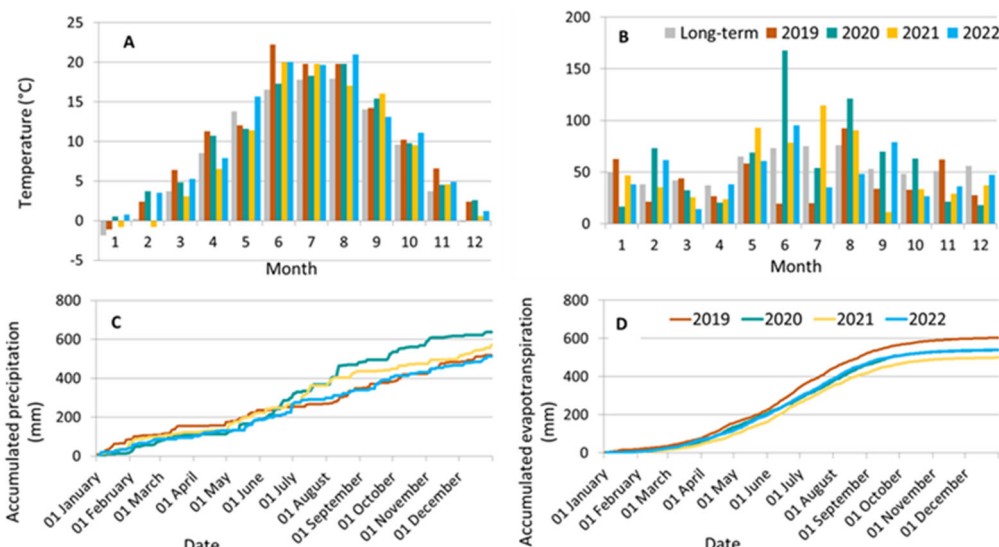

**Figure 1.** The month temperature (**A**) and month sums of precipitation (**B**); precipitation (**C**) and evapotranspiration (**D**) accumulated from January, in the years 2019–2022.

### 2.2. Layout of the Orchard and Experimental Irrigation Treatments

In this study, apple trees (*Malus domestica*) of the Red 'Jonaprince' variety, planted in the year 2013, were monitored in the years 2019–2022. The cultivation form is a slender spindle with a 'click' pruning modification. The trees were planted in a spacing of 3.5 × 1.2 m with a 1.5 m wide herbicide strip placed under the crown of the trees and the grass strips between the rows of tree. The rows of trees in the orchard are situated in a north-south direction and are parallel, always at the same distance from each other. The orchard was fertilised annually with the same dose of NPK fertiliser (16.5/16.5/16.5): 769 kg ha$^{-1}$. Half

of the dose was applied to the soil's surface on the herbicide strips in April, and the other half at the beginning of June.

The irrigation treatments ET100, ET75, and ET50 were examined, representing 100%, 75%, and 50% of the calculated actual evapotranspiration (ETc) value in April–September, respectively (Figure 2). A non-irrigated ET0 treatment served as the rain-fed control. These treatments were situated in the part of the orchard that was not protected by a net.

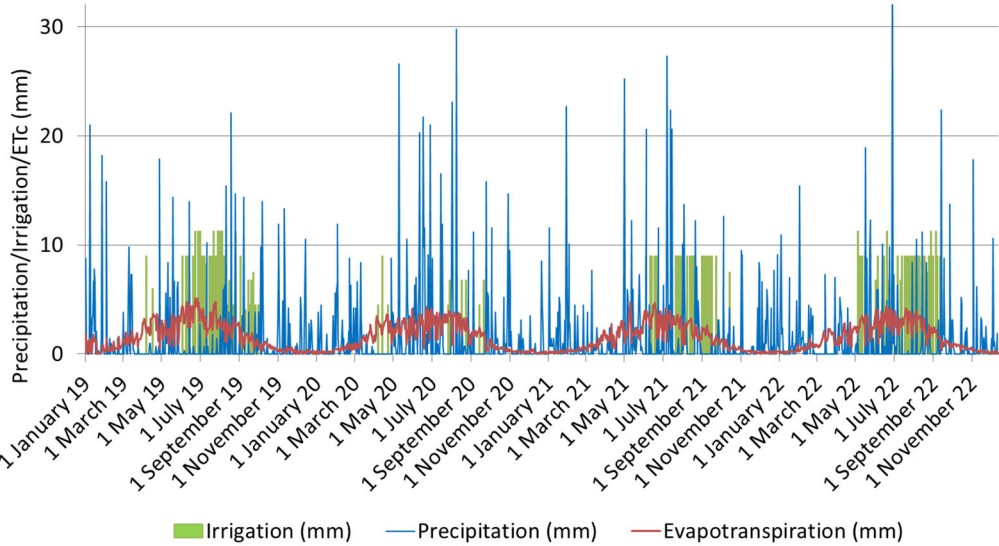

**Figure 2.** Daily precipitation sums and calculated actual evapotranspiration (ETc). Irrigation doses in treatment ET100 are presented.

The irrigation doses were calculated as the balance difference between the ETc and the observed amount of precipitation. The distribution and rates of irrigation during the vegetation period in the experimental years depended on the factors of temperature, evapotranspiration, and distribution of precipitation during the year. The ETc was calculated using the agrometeorological model AVISO ("Agrometeorological Computing and Information System") used in the Czech Agrometeorological Institute for water balance calculations for various crops and fruit trees. The model is primarily intended for the analysis of cases with a prevailing precipitation deficit. Evapotranspiration is calculated in a daily step using a modified procedure according to the Penman-Montheith algorithms [23–25]. The daily data for weather characteristics obtained from a meteorological station located in the experimental orchard were used for calculation of ETc. The soil moisture data were continuously monitored using Virrib sensors (Amet, Velké Bílovice, Czech Republic) [8]. The sensor is based on the TDR method and consists of two stainless-steel concentric circles placed horizontally into a soil depth of 30 cm. The soil moisture data obtained from the sensors during treatments ET100, ET50, and ET0 are presented in Figure 3. The available soil water content (ASWC) was expressed as the ratio of actual ASWC to the maximum ASWC at field capacity.

The irrigation treatments were performed in sections consisting of 17 trees in a particular row, randomly determined at the beginning of the experiment (this experiment also included other variants—sections of rows and neighbouring rows for which $^{13}$C analysis was not performed). Irrigation drip lines were guided along one side of the tree trunk at a height of 40 cm. The distance between the water emitters on the drip lines was 50 cm and the emitter capacity was 2.3 L per hour. Irrigation was conducted from April to September, on the same days for all treatments, in small doses of 6–9 mm, 2 to 3 times per week, to ensure the uniform absorption of water into the soil and to prevent possible seepage beyond the root zone.

The total water received by the apple tree during the ET100 treatment in 2019 was 548 mm (298 mm irrigation + 250 mm rainfall); in 2020, it was 556 mm (54 mm irrigation

+ 502 mm rainfall); in 2021, it was 606 mm (196 mm irrigation + 410 mm rainfall); and in 2022, it was 616 mm (273 mm irrigation + 343 mm rainfall). The ET50 and ET75 treatments received 50% and 75%, respectively, of the applied irrigation rates of ET100.

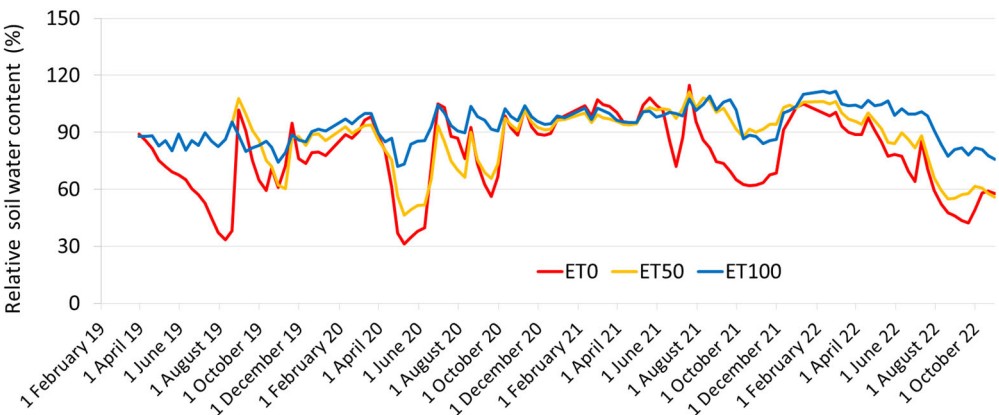

**Figure 3.** The course of relative available soil water content (actual SWC/ASWC at FWC) based on the data collected from the sensors during treatments ET0, ET50, and ET100 in the experimental years.

### 2.3. Terms of Sampling of Apple Leaves

### 2.3.1. Experiment A

Leaves with petioles were sampled in three replications, from approximately half the height of the tree crown in spring (June) and summer (September). At least 10 leaves were obtained for each replication. The leaves from both sides of a row were collected. The sampling terms in the experimental years were 3 (2019), 17 (2020), 5 (2021), and 6 (2022) June, and 12 (2019), 12 (2020), 16 (2021), and 2 (2022) September. The spring and summer samplings were conducted during the developmental stages BBCH 70–72 and BBCH 81–83, respectively.

### 2.3.2. Experiment B

The leaves of five classes, corresponding to their positions on the branches, and current year shoots were sampled for the ET0, ET50, and ET100 treatments. The sampling was performed in autumn (BBCH 90–91), on 11 November 2019, 26 October 2020, 10 November 2021, and 11 November 2022. Leaves growing from branches that were two of more years and one year old were designated L1 and L2, respectively. The leaves growing at the base, middle, and top of the shoots growing in the current year were labelled, in order, from L3 to L5, respectively (Figure 4). Leaves were sampled in two replications. At least 10 leaves were obtained from the respective positions from at least five trees. The leaves were sampled from branches on the same side (East) of the tree rows.

### 2.4. Chemical Analysis of the Leaves

Chemical Analysis

The whole leaves (including petioles and veins) obtained from experiments A and B were dried and homogenised into a fine powder in an MM301 ball mill (Retsch, Haan, Germany). The ratio of $^{13}C$ and $^{12}C$, $^{15}N$ and $^{14}N$, and the contents of total N and C were determined on an elemental analyser coupled to an isotope-ratio mass spectrometer EA Vario PYRO cube (Elementar, Langenselbold, Germany) with IRMS Isoprime precisION (Elementar, Manchester, UK). The values of $\delta^{13}C$, $\Delta^{13}C$, and $\delta^{15}N$ were calculated [10,11,26].

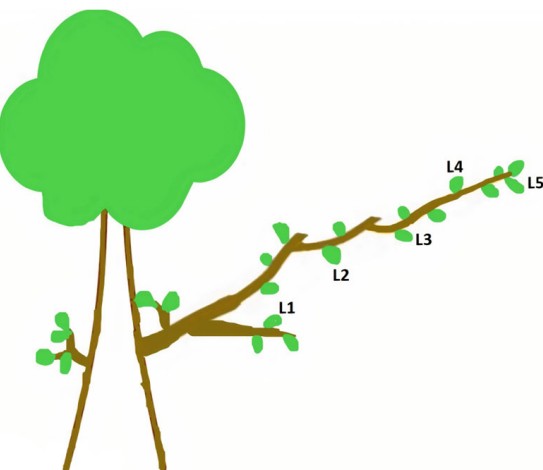

**Figure 4.** The scheme of branches and shoots where the leaves obtained from different positions on the tree, from L1 to L5, were sampled.

### 2.5. Statistical Analysis

The effects of water treatment, year, and the term of sampling (spring, summer) on the $\Delta^{13}C$ values were examined using two- (pooled data from two sampling terms) or three-way analysis of variance (ANOVA) tests. The $\Delta^{13}C$ data had a normal distribution. The differences among the mean values were evaluated using Tukey's HSD test (at $p < 0.05$), where ANOVA presented a significant effect. Linear regression analysis was also performed, and Pearson's correlation coefficient (r) was calculated for the characteristics under examination. STATISTICA 14 program (TIBCO software, StatSoft, Inc., Tulsa, OK, USA) was also used.

### 3. Results

In the four-year experiment, the effect of irrigation rates on the $^{13}C$ discrimination ($\Delta^{13}C$) of apple leaves obtained from apple trees was studied. The leaves were collected in the months of June (spring) and early September (summer) (Experiment A) and in autumn, November (Experiment B).

### 3.1. The Effect of Irrigation on $\Delta^{13}C$—Experiment A

Irrigation rates had a statistically significant impact on the $\Delta^{13}C$ values when the data obtained for spring and summer were pooled ($p = 0.002$) (Table 2). On average, the $\Delta^{13}C$ of leaves increased from 20.77‰ and 20.73‰ in ET0 and ET50 to 20.80‰ and 20.95‰ in ET75 and ET100, respectively. The average $\Delta^{13}C$ values for ET0 and ET50 were significantly different from ET100. However, within a year, the values for $\Delta^{13}C$ were not significantly different among the treatments.

When the $\Delta^{13}C$ data collected from the two sampling terms were analysed separately in the study, the effect of irrigation was marginally significant ($p = 0.039$ and $p = 0.043$) (Table 3). During the individual experimental years and terms, $\Delta^{13}C$ was always higher in treatment ET100 than in ET0; however, the differences were mostly minor (especially in spring 2019 and 2022, and summer 2021 and 2022) and insignificant.

The analysis showed a strong effect of the year on leaf $\Delta^{13}C$. The effect of the conditions experienced during the year was apparent in all treatments and in both terms (Tables 2 and 3). On average of treatments and sampling terms, the highest $\Delta^{13}C$ value, 21.29‰, was observed in 2021, and the lowest, 20.45‰, in 2020.

The year-to-year variability of $\Delta^{13}C$ did not show a dependence on the irrigation intensity. The coefficients of variance for the $\Delta^{13}C$ year data were 1.59%, 2.01%, 1.62%, and 1.89% from ET0 through ET100, respectively, in spring, The corresponding summer coefficients of variance were 1.51%, 1.02%, 1.40%, and 1.24%.

The difference between ET0 and ET100 was greater in summer than in spring in 2019 (−0.02‰ against −0.27‰) and 2020 (−0.09‰ against −0.46‰), which means the $\Delta^{13}C$ increased more in ET100 than in ET0, between spring and summer. In the rainy year of 2021, the difference changed conversely (−0.30‰ against −0.16‰) and a nil effect was observed in 2022.

The effect of irrigation on fruit yields was not significant [27]. Average yields for variants ET0, ET 50, ET75, and ET100 for the years 2019–2022 were 26.48, 29.85, 29.18, and 26.15 kg tree$^{-1}$. That means WUE decreased with increasing irrigation rates. In individual years, yields fluctuated slightly, for example, in 2022, yields of 26.02, 29.90, 25.60, and 23.21 kg tree$^{-1}$ were recorded for variants ET0, ET 50, ET75, and ET100, respectively.

**Table 2.** The analysis of variance for the effect of irrigation treatments, year, and the term of sampling on the $\Delta^{13}C$ of apple leaves. The means within the same factor or the factor's interaction followed by different letters are significantly different ($p < 0.05$).

| Factor | $p$ | Factor Level | | $\Delta^{13}C$ ‰ | Std. Deviation ‰ | |
|---|---|---|---|---|---|---|
| Treatment (Treat) | 0.002 | ET0 | | 20.77 | 0.38 | b |
| | | ET50 | | 20.73 | 0.37 | b |
| | | ET75 | | 20.80 | 0.36 | ab |
| | | ET100 | | 20.95 | 0.38 | a |
| Term | 0.867 | Spring | | 20.82 | 0.40 | |
| | | Summer | | 20.81 | 0.35 | |
| Year | <0.001 | 2019 | | 20.80 | 0.21 | c |
| | | 2020 | | 20.45 | 0.27 | b |
| | | 2021 | | 21.29 | 0.19 | a |
| | | 2022 | | 20.72 | 0.22 | c |
| Average | | | | 20.81 | 0.38 | |
| Year × Term | <0.001 | 2019 | Spring | 20.86 | 0.20 | b |
| | | 2019 | Summer | 20.73 | 0.21 | b |
| | | 2020 | Spring | 20.30 | 0.17 | c |
| | | 2020 | Summer | 20.61 | 0.27 | b |
| | | 2021 | Spring | 21.32 | 0.18 | a |
| | | 2021 | Summer | 21.26 | 0.21 | a |
| | | 2022 | Spring | 20.79 | 0.13 | b |
| | | 2022 | Summer | 20.64 | 0.27 | b |
| Year × Treat | 0.801 | 2019 | ET0 | 20.80 | 0.24 | |
| | | 2019 | ET50 | 20.64 | 0.10 | |
| | | 2019 | ET75 | 20.81 | 0.23 | |
| | | 2019 | ET100 | 20.95 | 0.15 | |
| | | 2020 | ET0 | 20.36 | 0.28 | |
| | | 2020 | ET50 | 20.44 | 0.27 | |
| | | 2020 | ET75 | 20.39 | 0.16 | |
| | | 2020 | ET100 | 20.63 | 0.33 | |
| | | 2021 | ET0 | 21.23 | 0.15 | |
| | | 2021 | ET50 | 21.23 | 0.27 | |
| | | 2021 | ET75 | 21.24 | 0.14 | |
| | | 2021 | ET100 | 21.46 | 0.12 | |
| | | 2022 | ET0 | 20.72 | 0.22 | |
| | | 2022 | ET50 | 20.61 | 0.19 | |
| | | 2022 | ET75 | 20.76 | 0.27 | |
| | | 2022 | ET100 | 20.78 | 0.20 | |
| Treat × Term | 0.675 | | | | | |
| Treat × Term × Year | 0.495 | | | | | |

### 3.2. The Effect of Irrigation on $\Delta^{13}C$—Experiment B

The significant effect of a leaf's position on the average $\Delta^{13}C$ value was observed in Experiment B ($p < 0.001$) (Table 4). The average values of $\Delta^{13}C$ decreased from the L1 leaves to youngest ones at the shoot apex (L5), and were 22.44‰, 21.74‰, 22.05‰, 21.37‰, and 21.00‰, respectively. The course of the decrease in $\Delta^{13}C$ from L1 through L5 was similar in the years 2019–2021 and slightly different in 2022 (Figure 5). The values of $\Delta^{13}C$ for L1 and L2 leaves were significantly higher than those for L4 and L5 in all the experimental years.

**Table 3.** The analysis of variance for the effect of irrigation treatments and year on the $\Delta^{13}C$ of apple leaves for spring and summer terms of sampling. The means within the same factor or factor's interaction followed by different letters are significantly different ($p < 0.05$).

| | | **Spring** | | | | **Summer** | | | | | |
|---|---|---|---|---|---|---|---|---|---|---|---|
| **Factor** | ***p*** | **Factor Level** | **$\Delta^{13}C$** ‰ | **Std. Deviation** ‰ | | **Factor** | ***p*** | **Factor Level** | **$\Delta^{13}C$** ‰ | **Std. Deviation** ‰ | |
| Treatment (Treat) | 0.039 | ET0 | 20.80 | 0.39 | ab | Treatment (Treat) | 0.043 | ET0 | 20.75 | 0.39 | a |
| | | ET50 | 20.72 | 0.45 | b | | | ET50 | 20.73 | 0.29 | a |
| | | ET75 | 20.83 | 0.39 | ab | | | ET75 | 20.78 | 0.36 | a |
| | | ET100 | 20.92 | 0.42 | a | | | ET100 | 20.99 | 0.34 | a |
| Year | <0.001 | 2019 | 20.86 | 0.20 | b | Year | <0.001 | 2019 | 20.86 | 0.20 | b |
| | | 2020 | 20.30 | 0.17 | c | | | 2020 | 20.30 | 0.17 | b |
| | | 2021 | 21.32 | 0.18 | a | | | 2021 | 21.32 | 0.18 | a |
| | | 2022 | 20.79 | 0.13 | b | | | 2022 | 20.79 | 0.13 | b |
| Average | | | 20.82 | 0.40 | | Average | | | 20.81 | 0.35 | |
| Year × Treat | 0.289 | 2019 ET0 | 20.96 | 0.03 | | Year × Treat | 0.835 | 2019 ET0 | 20.63 | 0.26 | |
| | | 2019 ET50 | 20.66 | 0.15 | | | | 2019 ET50 | 20.61 | 0.06 | |
| | | 2019 ET75 | 20.85 | 0.27 | | | | 2019 ET75 | 20.78 | 0.23 | |
| | | 2019 ET100 | 20.99 | 0.16 | | | | 2019 ET100 | 20.91 | 0.17 | |
| | | 2020 ET0 | 20.29 | 0.36 | | | | 2020 ET0 | 20.42 | 0.23 | |
| | | 2020 ET50 | 20.21 | 0.08 | | | | 2020 ET50 | 20.67 | 0.14 | |
| | | 2020 ET75 | 20.31 | 0.08 | | | | 2020 ET75 | 20.48 | 0.19 | |
| | | 2020 ET100 | 20.38 | 0.02 | | | | 2020 ET100 | 20.88 | 0.30 | |
| | | 2021 ET0 | 21.19 | 0.18 | | | | 2021 ET0 | 21.27 | 0.15 | |
| | | 2021 ET50 | 21.37 | 0.06 | | | | 2021 ET50 | 21.10 | 0.35 | |
| | | 2021 ET75 | 21.24 | 0.21 | | | | 2021 ET75 | 21.24 | 0.06 | |
| | | 2021 ET100 | 21.49 | 0.14 | | | | 2021 ET100 | 21.42 | 0.11 | |
| | | 2022 ET0 | 20.76 | 0.03 | | | | 2022 ET0 | 20.67 | 0.34 | |
| | | 2022 ET50 | 20.66 | 0.15 | | | | 2022 ET50 | 20.56 | 0.26 | |
| | | 2022 ET75 | 20.92 | 0.10 | | | | 2022 ET75 | 20.61 | 0.32 | |
| | | 2022 ET100 | 20.82 | 0.01 | | | | 2022 ET100 | 20.74 | 0.31 | |

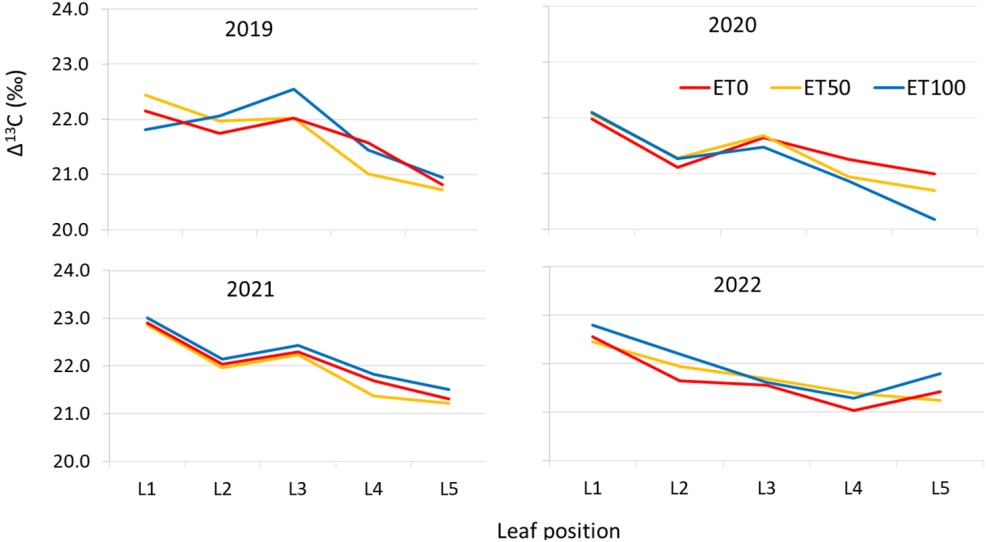

**Figure 5.** The effect of irrigation treatments on values of $\Delta^{13}C$ at successive positions on apple shoots.

**Table 4.** The analysis of variance for the effect of irrigation treatments, year, and the position of leaves on the $\Delta^{13}$C of apple leaves (ANOVA). The means within the same factor or the factor's interaction followed by different letters are significantly different ($p < 0.05$).

| Factor | *p* | Factor Level | | $\Delta^{13}$C ‰ | Std. Deviation ‰ | |
|---|---|---|---|---|---|---|
| Year | <0.001 | 2019 | | 21.74 | 0.66 | b |
| | | 2020 | | 21.30 | 0.57 | c |
| | | 2021 | | 22.05 | 0.59 | a |
| | | 2022 | | 21.78 | 0.54 | b |
| Treatment (Treat) | 0.022 | ET0 | | 21.69 | 0.56 | b |
| | | ET50 | | 21.66 | 0.63 | b |
| | | ET100 | | 21.81 | 0.73 | a |
| Leaf position | <0.001 | L1 | | 22.50 | 0.40 | a |
| | | L2 | | 21.78 | 0.41 | b |
| | | L3 | | 21.94 | 0.39 | b |
| | | L4 | | 21.31 | 0.36 | c |
| | | L5 | | 21.07 | 0.47 | d |
| Average | | | | 21.72 | 0.64 | |
| Year × Leaf | 0.001 | 2019 | L1 | 22.44 | 0.31 | ab |
| | | 2019 | L2 | 21.92 | 0.24 | bcd |
| | | 2019 | L3 | 22.20 | 0.31 | b |
| | | 2019 | L4 | 21.34 | 0.26 | ef |
| | | 2019 | L5 | 20.83 | 0.22 | fg |
| | | 2020 | L1 | 22.05 | 0.20 | bc |
| | | 2020 | L2 | 21.22 | 0.27 | ef |
| | | 2020 | L3 | 21.60 | 0.14 | de |
| | | 2020 | L4 | 21.02 | 0.33 | f |
| | | 2020 | L5 | 20.62 | 0.38 | g |
| | | 2021 | L1 | 22.93 | 0.23 | a |
| | | 2021 | L2 | 22.05 | 0.17 | bc |
| | | 2021 | L3 | 22.32 | 0.09 | b |
| | | 2021 | L4 | 21.63 | 0.21 | de |
| | | 2021 | L5 | 21.34 | 0.14 | ef |
| | | 2022 | L1 | 22.60 | 0.15 | ab |
| | | 2022 | L2 | 21.93 | 0.23 | bcd |
| | | 2022 | L3 | 21.62 | 0.17 | de |
| | | 2022 | L4 | 21.23 | 0.27 | ef |
| | | 2022 | L5 | 21.49 | 0.33 | de |
| Treat × Leaf | 0.409 | ET0 | L1 | 22.40 | 0.44 | |
| | | ET0 | L2 | 21.63 | 0.40 | |
| | | ET0 | L3 | 21.88 | 0.33 | |
| | | ET0 | L4 | 21.39 | 0.34 | |
| | | ET0 | L5 | 21.14 | 0.35 | |
| | | ET50 | L1 | 22.46 | 0.37 | |
| | | ET50 | L2 | 21.79 | 0.38 | |
| | | ET50 | L3 | 21.91 | 0.31 | |
| | | ET50 | L4 | 21.18 | 0.31 | |
| | | ET50 | L5 | 20.97 | 0.31 | |
| | | ET100 | L1 | 22.66 | 0.39 | |
| | | ET100 | L2 | 21.91 | 0.45 | |
| | | ET100 | L3 | 22.02 | 0.53 | |
| | | ET100 | L4 | 21.35 | 0.42 | |
| | | ET100 | L5 | 21.10 | 0.69 | |
| Year × Treat | 0.013 | | | | | |
| 3-way interaction | 0.617 | | | | | |

The values of $\Delta^{13}$C were significantly affected by the experimental year ($p < 0.001$), with the highest average values presented in 2021 and the lowest in 2020, similar to the leaves sampled in spring and summer.

The average values of $\Delta^{13}$C were significantly higher in ET100 than in ET0; however, during the individual years, the effect was significant only in the year 2019. At the level of individual leaves' positions, the $\Delta^{13}$C value of L1 for all three irrigation treatments was significantly higher than in the other positions, and the $\Delta^{13}$C values for L1, L2, and L3 were significantly higher than the $\Delta^{13}$C values for L4 and L5, also in all treatments (Table 4).

Similar to the spring and summer samplings, the comparison of $\Delta^{13}$C in ET0 and ET100 did not show that the highest doses of irrigation (ET100) balanced the $\Delta^{13}$C differences among years. The coefficient of variance for $\Delta^{13}$C in the experimental years was higher in all leaf positions in ET100 (on average 1.99%) than in ET50 (1.17%) and ET0 (1.37%).

### 3.3. The Relationships between $\Delta^{13}$C and the Contents of N and C and $\delta^{15}$N of Apple Leaves

In Experiment A, the $^{13}$C discrimination of apple leaves in all experimental years presented a significant relation to the contents of N ($r = 0.76$, $p < 0.001$, N = 32) and C ($r = 0.53$, $p = 0.002$, N = 32) in the leaves. The scatter plots suggest different slopes of the regression line in the experimental years, especially for $\Delta^{13}$C and C or N contents (Figure 6). The plots also present an apparent divergence of $\delta^{15}$N values in 2019 from the other years.

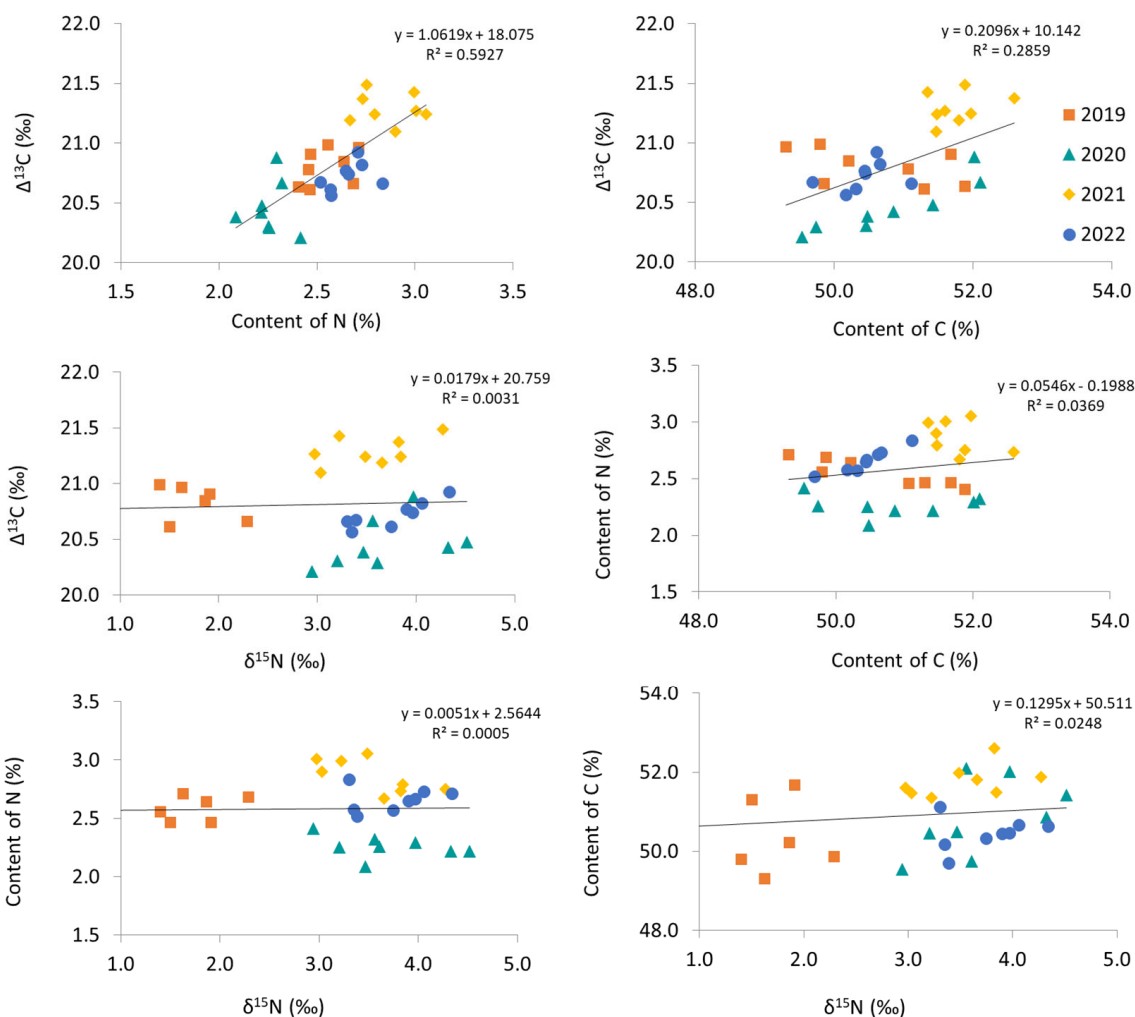

**Figure 6.** The relationships between $\Delta^{13}$C, $\delta^{15}$N, and the contents of N and C of apple leaves. The data for both spring and summer sampling terms (Experiment A) are presented.

In Experiment B, the correlation analysis of the data obtained from all the experimental years showed significant negative relationships between $\Delta^{13}C$ and $\delta^{15}N$ (r = −0.61, $p < 0.001$, N = 60) and between $\Delta^{13}C$ and C content (r = −0.41, $p = 0.001$, N = 60). The correlations between the values of $\Delta^{13}C$, $\delta^{15}N$, and the contents of N and C of apple leaves were different from Experiment A due to a strong relationship between the leaf position and the data of analysis.

When averaged by year, the values of $\Delta^{13}C$ decreased from leaf L1 to L5, while N and C contents increased until L4 and decreased for L5 (Figure 7). The average values of $\delta^{15}N$ increased from L1 through to L5: 2.12‰, 2.21‰, 2.38‰, 2.84‰, and 3.76‰, respectively. The difference was the greatest in 2020, when $\delta^{15}N$ reached 5.33–5.66‰ in the youngest leaves (L5), 3.42–3.94‰ in L4, and 2.17–2.73‰ in the others leaves. The N content increased to L4 (1.91%) but decreased again in the youngest leaves L5 (1.79%), to the levels of L1–L3 (1.76–1.80%). Similarly, the C content increased from L1 to L4 (47.75 to 50.35%) and decreased again in L5 (49.79%).

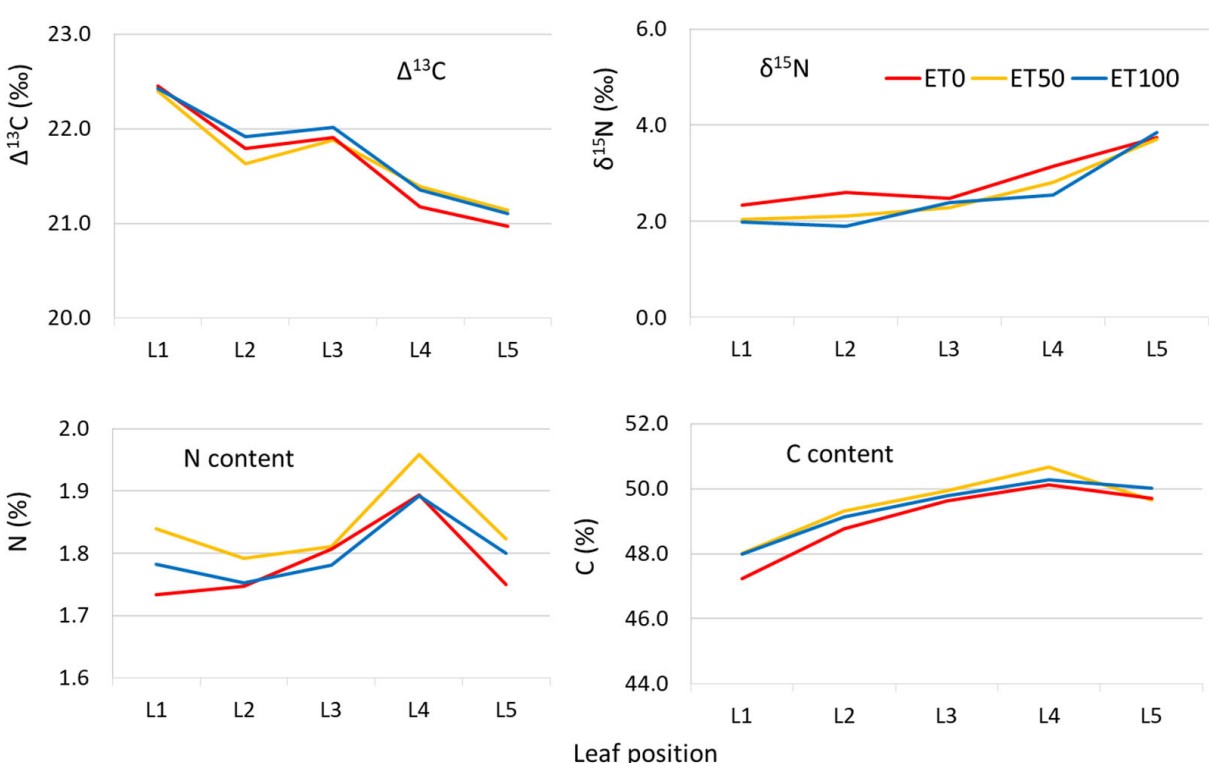

**Figure 7.** The values of $\Delta^{13}C$, $\delta^{15}N$, and the contents of N and C of the apple leaves averaged over years in Experiment B.

## 4. Discussion

### 4.1. The Effect of Irrigation Rates on $\Delta^{13}C$

The four-year study presented here was aimed at obtaining the $^{13}C$ signature of apple leaves under increasing irrigation rates. We assumed that the $^{13}C$ discrimination of apple leaves would differentiate among treatments using different water supplies.

The effect of irrigation on $\Delta^{13}C$ was confirmed when the data collected from spring and summer were pooled; however, the effect was weak in the spring and summer data ($p < 0.043$) and the analysis did not substantiate the significant effect of irrigation in the individual years (Tables 2 and 3). It should be noted that $\Delta^{13}C$ was higher in ET100 in comparison to ET0 in all years and terms of sampling; however, the differences were mostly minor, except for the summer months in 2019 and 2020 (0.27‰, 0.46‰) and spring 2021 (0.30‰), and insignificant. Even the greatest differences between the non-irrigated and fully irrigated treatments we observed were minor, in comparison to the data collected

from irrigated and rain-fed or stressed crops [12,28] and fruits [13,29]. The weak effect of increasing irrigation doses on $\Delta^{13}C$ corresponded to an insignificant effect on fruit yield in this experiment. However, unlike $^{13}C$ discrimination fruit yields were the lowest in ET0 and ET100 [27]. It suggests the yield formation is affected by a complex of factors while the $\Delta^{13}C$ increased from ET0 to ET100.

According to the theory and published experimental data in the literature, the shortage of water increases plant $^{13}C$ content (the value of $\Delta^{13}C$ decreases). The relationship is so strong and robust that it has been used to reconstruct moisture variability a century or more ago [30,31]. Commonly, in crops growing under water stress conditions, for example, in (semi)arid areas or under induced water shortage, the values of $\Delta^{13}C$, of approximately 13–18‰, were reported in the studies of Wahbi and Shaaban [32], Flohr et al. [31], and Dalal et al. [33]. On the other hand, for crops growing with a sufficient supply of water or additional irrigation sources, the $\Delta^{13}C$ value exceeded 20–22‰ [28]. However, the results obtained by Flohr et al. [31] show a large variation in the carbon stable isotope values of wheat and barley that received similar amounts of water, either as an absolute water input or as a percentage of crop requirements.

The apple tree data available in the literature on $\Delta^{13}C$ do not provide such clear, strong relationships between $\Delta^{13}C$ and irrigation. Glenn [6,15] reported the leaf $\Delta^{13}C$ values of orchard apples in the range 18.8–20.5‰ and 18.8–25.5‰, respectively. The author did not observe a clear effect of irrigation (70% of pan evaporation) on leaf and shoot $\Delta^{13}C$ values in studies conducted on 'Empire' apples [6,15]. Biasuz et al. [14] reported a rather low range of $\Delta^{13}C$ values, 17–19‰, for 'Honeycrisp' apple leaves under irrigation replacing 110% of the estimated evapotranspiration levels during the growing season. In our experiment, the $\Delta^{13}C$ fell under 20.5‰ only in ET0, ET50, and ET75 samples in spring 2020. The highest values of $\Delta^{13}C$, 21.10–21.49‰, were observed during both spring and autumn sampling terms in 2021. The relatively high levels of $\Delta^{13}C$ suggest the lower sensitivity of apple trees to water shortages and/or a sufficient amount of available water even in ET0. This result was also documented by the very small difference between non-irrigated ET0 (on average 20.77‰) and ET50 (20.73‰), irrigated with 50% of water lost by evapotranspiration.

The level of reduced soil water availability having a significant impact on the growth and yields of crops is usually determined at approximately 40–60% of the available soil water capacity [7,34]; the irrigation of fruit trees should begin when the soil moisture level in the effective root zone decreases to less than 40% and 50% of the available water capacity (AWC). Evidently, numerous factors affect the real impact of certain levels of available water content [31]. In the present experiment, trees in ET0 grew under 50% of AWC for only a small portion of the vegetation period in all years. For the majority of the growing season, trees grew under conditions of higher water content (Figure 3) that contributed little to $\Delta^{13}C$.

The experimental data did not conclusively confirm the assumption that the irrigation during the growth period increased $\Delta^{13}C$ more in ET100 than in ET0. As a result, the difference between ET0 and ET100 samples should increase when comparing the $\Delta^{13}C$ values for spring and summer, especially during dry years. The assumption could be confirmed in the dry year of 2019 and also in the year 2020, which experienced more rain, with the lowest $\Delta^{13}C$ value (Table 4). In 2021, the year with the highest level of $\Delta^{13}C$ for all experimental years, the difference between the sampling terms was minimal in both treatments, and in 2022 (with similar precipitation levels as in 2019), the difference between ET0 and ET100 was the same in spring and summer, which might be attributed to the slow decrease in soil moisture levels in both ET0 and ET100 treatments. These results document, again, that the effect of irrigation on the $\Delta^{13}C$ in the experiment, on the edge of evidence.

### 4.2. The Year Variability of $\Delta^{13}C$

The conditions experienced during the year had a strong significant effect on $\Delta^{13}C$ in both sampling terms; however, a clear consistent relationship between $\Delta^{13}C$ and precipitation or soil moisture levels during the growth period, which could be expected, at least, in

　　　　　　　　　　　　　　　　　　　　　　　　　　　　　　　　

the non-irrigated control, was not evident. The precipitation was unevenly distributed during the vegetation months, which caused a fluctuation in soil water availability (Figure 2) and complicated our interpretation of the results.

Additionally, the experimental data did not present the clear effect of irrigation on the variability of $\Delta^{13}C$ during the treatments. Thus, the assumption that irrigation balances the year variability of $\Delta^{13}C$ could not be confirmed. The same conclusion can be determined from the data presented in Experiment B.

Soil water availability monitored using moisture sensors corresponded slightly better to the observed year variability of $\Delta^{13}C$ in ET0 than the precipitation sums, as shown in the results obtained for the year 2020 (Figures 1 and 3). Cumulative precipitation values, from the month of January, presented a similar input value of rainwater during all years until approximately mid-June, when stronger rainfall activity occurred in 2020. Therefore, a low water requirement was calculated in 2020 and irrigation levels were the lowest (54 mm) in the experimental years; however, the soil moisture levels in ET0 and ET50 rapidly decreased, and the decrease was also monitored in ET100 (Figure 2). This probably contributed to the low $\Delta^{13}C$ value and increasing difference of $\Delta^{13}C$ between ET0 and ET100 in the spring and summer sampling terms, as previously discussed. Some discrepancies between the precipitation sums and sensor data may be the result of soil spatial variability, the preferential flow of water, and water root uptake distribution [35–37]. Moisture sensors were placed in the soil at a depth of 30 cm (where the main effect of small irrigation doses on soil moisture was expected), while the root system of the apple trees reached down to 80 cm in the experiment [38] and could provide access to an additional water reserve during episodes without significant precipitation activity.

### 4.3. The Indices for the Use of Carbon from the Previous Year

The effect of irrigation on the $\Delta^{13}C$ of leaves was questioned by the fact that differences among the treatments were observed, not only in summer but also in spring, prior to the initiation of the irrigation process. Furthermore, the effect of years on $\Delta^{13}C$ during the years were similar during the zero-irrigation treatment (ET0) and for trees with a maximal water supply (ET100).

The values of $\Delta^{13}C$ integrate the reaction of plants on water availability over a long period [15,39], which could be the reason for the ambiguous results. Several authors showed, using $^{14}CO_2$ or $^{13}CO_2$ exposure for apple tree during growth, the storage and utilisation of C in the next season. The importance of roots when storing carbohydrate reserves for the following season was stressed by Breen et al. [40]. According to Loescher et al. [41], root reserves increase late in the growing season, and the accumulation of these reserves is therefore very sensitive to late-season stresses. Conversely, in a pot experiment, Imada and Tako [42] observed that the concentration of assimilated $^{13}C$ in late autumn, in the woody parts, was higher in the trees labelled during the period of vigorous vegetative growth, compared to those labelled during other growth periods. The use of C assimilated and stored during the hot and dry year of 2019 may explain the low $\Delta^{13}C$ in 2020 already during the spring sampling date. In this context, it may be noted that also the year 2018 was extremely dry and hot, with precipitation levels reaching 55% of the long-term average. Furthermore, the total root length (corresponding to the mass) of apple trees in the same irrigation experiment was the longest in 2020 [38], probably in response to the water shortage that occurred in 2019, and the low precipitation levels during winter and at the start of 2020 that were insufficient to refill water reserves in deeper subsoil reserves. The soil moisture level was approximately 80–90% of AWC for most of the vegetation, even in fully irrigated ET100 treatment in 2019, in contrast to the other years (Figure 3). Nonetheless, it remains difficult to explain why the $\Delta^{13}C$ in spring 2020 was lower during all treatments, including ET100, than in autumn 2019. This highlights the possibility that, during dry years, some parts of the root system, in over-dried soil, deeper or located horizontally further from soil wetted by drippers, may signal water shortage conditions, similar to systems of regulated deficit irrigation or partial root-zone drying [43]. The confirmation of this hypothesis would

demand the extensive monitoring of the water status of the root zone and leaves during the growth of apple trees in the orchard, which was not the objective of this study.

### 4.4. The Effect of Leaf Position (Experiment B)

We assumed that the utilisation of C formed in the previous year could be reflected in the $^{13}$C discrimination of leaves in different positions on a tree corresponding to their respective age. Older leaves, growing from one-or-more-year-old branches probably utilise more carbon from their reserves than the youngest leaves located at the apex of annual shoots, which can use freshly assimilated C [41,44]. Our experimental data confirm the significant effect of positions of apple leaves on $\Delta^{13}$C. The values of $\Delta^{13}$C decreased consistently in all years, from the position L1 to the youngest leaves positioned at the apexes of annual (present year) shoots (position L5).

The average $\Delta^{13}$C values were the lowest in 2020 and the highest in 2021 (as in Experiment A); however, the decline in values from the older to youngest leaves was similar to the other years (Figure 5). The results suggest the effect of other physiological processes. For example, Vogado et al. [45] confirmed that emerging leaves (in deciduous and evergreen species) are initially $^{13}$C enriched compared to mature leaves growing on the same plant, with their $\delta^{13}$C decreasing during the leaf-expansion process. This is in agreement with our data; however, to the best of our knowledge, no data on these apples exist in the literature to confront our results. In annual crops, the interpretation of $^{13}$C data is more straightforward; for example, Dercon et al. [46] concluded that $\Delta^{13}$C values measured in different maize plant parts during harvest can be used as a historical account of how water availability varies during the entire cropping cycle. However, the effect of the position and age of the leaves does not explain the higher values of $\Delta^{13}$C in leaves in autumn compared to the data available for spring and autumn leaves.

The difference in $\Delta^{13}$C between contrasting variants ET100 and ET0 was expected to increase with leaf order due to irrigation exposure during the growth period. The results confirmed this assumption only in 2020; however, in the opposite direction, as the difference increased from 0.12‰ to −0.82‰, i.e., $\Delta^{13}$C was higher in ET100 than in ET0 in L1 and significantly lower in L5. In other years, the differences decreased (in 2019, from 0.56 to 0.13) or were minor.

### 4.5. The Relationships among $\Delta^{13}$C, $\delta^{15}$N, and N and C Contents of Apple Leaves

The discussion presented above shows the complexity of interpreting the $\Delta^{13}$C data. Other characteristics/analytical data determined for the same leaf samples, $\delta^{15}$N, N and C contents, could possibly contribute to the description of the conditions in individual years.

The $\Delta^{13}$C correlated positively with the content of N in leaves in Experiment A (Figure 6), which suggests that favourable conditions for the uptake and/or utilisation of nitrogen were also favourable for sufficient water uptake. In crops, negative relationships between $\Delta^{13}$C and N fertilisation were observed as the result of faster water depletion by fertilised crops with higher LAI [19,47]. Brillante et al. [18] observed a significant correlation between the $\Delta^{13}$C of grape must and leaf nitrogen, and they stated that it was unlikely that the effect of N could be strong enough to prevent a direct interpretation of $\Delta^{13}$C in terms of plant water stress. Other studies suggest a possible effect of the high sink (fruits) on photosynthesis and leaf nitrogen [48].

Whole-plant and leaf nitrogen isotope compositions are determined in the research by the isotope ratio of the external nitrogen source and physiological mechanisms occurring within the plant [49]. In our experiment, the value of $\delta^{15}$N in leaves was different in 2019, in comparison with the other years (Figure 6). Low values of $\delta^{15}$N in the dry year of 2019 could be related to a relatively low $N_{min}$ supply in 0–90 cm (91 kg N ha$^{-1}$) and, probably, to a greater proportion of N received from mineral fertilisers (having lower $\delta^{15}$N values nearer to air $\delta^{15}$N), while the conditions in the following years were more favourable for the mineralisation of N from soil organic matter and plant residues. Figure 6 also documents the different relationships between $\delta^{15}$N and $\Delta^{13}$C in the experimental years.

In Experiment B, the consistent effect of leaf position on $\Delta^{13}$C modified the direction of the relationships between $\Delta^{13}$C and $\delta^{15}$N, and N and C contents in comparison with Experiment A. The $\delta^{15}$N evident in the youngest leaves was consistently higher in all years, in comparison with the older ones; the difference was the greatest in 2020, when $\delta^{15}$N reached 5.59–6.13‰ in the youngest leaves (L5), 3.42–3.94‰ in L4, and 2.17–2.73‰ in the others leaves. This suggests a greater proportion of N received from the soil supply in young leaves; however, the functional explanation of the results should combine the effect of $N_{min}$, water availability, and distribution and deposition of N reserves in tree organs. To the best of our knowledge, the relationship between the $\delta^{15}$N and $\Delta^{13}$C discriminations of plants have not been studied in apples or other fruits. In ecophysiological, natural community studies, the strong significant effects of water availability on the relations between $\Delta^{13}$C and $\delta^{15}$N have been observed [50,51].

The aspect of genetic diversity adds to the difficulty of interpreting the abovementioned complex relationships. Plant reactions to different water availability levels often differ among cultivars due to numerous interacting factors and morphological and physiological traits. As a result, cultivar differences may be expected in the effect of irrigation on $\Delta^{13}$C [14,52,53]. The presented results for the four-year experiment were obtained from one apple cultivar, 'Red Jonaprince'; thus, more or less different results, compared to other cultivars and/or rootstocks, are also probable.

## 5. Conclusions

The results of the experiment confirm the significant effect of increasing irrigation rates on the $\Delta^{13}$C of leaves collected from apple trees. The relatively narrow range of observed $\Delta^{13}$C signatures and values over 20‰ suggest a low water-shortage level, even in non-irrigated, rain-fed apple trees. This probably contributed to a strong, significant effect of year conditions on the $\Delta^{13}$C in all irrigation treatments. Even the highest irrigation intensity, replenishing 100% of the calculated evapotranspiration losses, did not balance the year-to-year variability of leaf $\Delta^{13}$C. The data suggest that the use of carbon assimilated in the previous year affected $\Delta^{13}$C in the running year; however, the confirmation of this hypothesis would demand the extensive monitoring of soil and plant water status and the determination of the $\Delta^{13}$C of roots and other organs, not only during the growing season. The results indicate the continuity of processes over several years and the importance of water supply and agronomic measures sustaining the quality soil water traits during the long term than in only one growing season. Additionally, the strong consistent effect of the leaves' age on the $\Delta^{13}$C signature in all experimental years and the relationships of $\Delta^{13}$C to $\delta^{15}$N, N and C contents of leaves are worth exploring in the research conducted in the future.

**Author Contributions:** Conceptualisation, J.H. and I.R.; methodology, J.H., M.M. (Martin Mészáros) and I.R.; software, J.H.; validation, M.M. (Michal Moulik) and I.R.; formal analysis, M.M. (Michal Moulik), I.R. and M.M. (Martin Mészáros); investigation, I.R. and P.S.; resources, G.K.; writing—original draft preparation, J.H.; writing—review and editing, J.H., M.M. (Michal Moulik) and I.R.; visualisation, J.H.; supervision, G.K.; project administration, P.S., G.K.; funding acquisition, M.M. (Martin Mészáros). All authors have read and agreed to the published version of the manuscript.

**Funding:** This research was supported by the Ministry of Agriculture of the Czech Republic, project No. QK1910165 and institutional support MZE-RO0423.

**Data Availability Statement:** Data and scripts generated and/or analysed during this study are available from the corresponding author upon request.

**Acknowledgments:** The authors are thankful to Karina Kremleva for technical support.

**Conflicts of Interest:** The authors declare no conflict of interest.

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
