# Peer review of "The Effect of Increasing Irrigation Rates on the Carbon Isotope Discrimination of Apple Leaves"

_agronomy, doi:10.3390/agronomy13061623_

Round 1
Reviewer 1 Report
Comments and Suggestions for Authors
The paper titled ‘The effect of increasing irrigation rates on the carbon isotope 2 discrimination of apple leaves’ aims to examine the effect of increasing irrigation doses on the Δ13C of apple leaves. The Authors assumed that the Δ13C signature significantly distinguishes between treatments with irrigation from a rain-fed control, apple trees dependent only on precipitation. The manuscript is an intriguing contribution in the field of plant physiology and adaptive mechanisms to extreme weather conditions referring to 13C discrimination in plants’ leaves.
The scientific level of the publication is appropriate. The research was conducted and described thoroughly. The Authors achieved the set goals, however, I wonder in which way the results obtained, can be used? Are they to control irrigation or maybe to manage the water management of the catchment area? Or there is another utilitarian objective?
The abstract is informative, the manuscript is clearly and concisely written. The method are well described however, I would like the Authors to specify some issues:
− On what basis were the rows selected for the research?
− How were the rows of trees positioned relative to cardinal directions?
− Were all rows of tested trees parallel to each other?
− In which direction of the world were the branches from which the leaf samples were taken?
− Were samples taken from branches facing the same direction?
− The range of the root zone of apple trees is up to 80 cm down. Why the soil moisture changes were not monitored deeper than 30 cm?
The range of references is appropriate and properly cited. In overall, after some minor corrections done by the Authors I recommend the manuscript for being published in the journal Agronomy.
Author Response
Dear reviewer,
we appreciate the time and effort that you dedicated to providing feedback on our manuscript and are grateful for the insightful comments on and valuable improvements to our paper. Thanks for the thorough manuscript review.
The paper titled ‘The effect of increasing irrigation rates on the carbon isotope 2 discrimination ofapple leaves’ aims to examine the effect of increasing irrigation doses on the Δ13C of apple leaves. The Authors assumed that the Δ13C signature significantly distinguishes between treatments with irrigation from a rain-fed control, apple trees dependent only on precipitation. The manuscript is an intriguing contribution in the field of plant physiology and adaptive mechanisms to extreme weather conditions referring to 13C discrimination in plants’ leaves.
The scientific level of the publication is appropriate. The research was conducted and described thoroughly. The Authors achieved the set goals, however, I wonder in which way the results obtained, can be used? Are they to control irrigation or maybe to manage the water management of the catchment area? Or there is another utilitarian objective?
Answer: Thank you for carefully reading the manuscript and relevant comments. We understand the 13C data as summary information about water availability and tree water management, which integrates conditions over a longer period, so it is a retrospective assessment of conditions. We have
emphasized this aspect in the text. It is a different view than, for example, the course of the physical characteristics of water availability in a certain soil layer, which can be used for irrigation management (with the knowledge of other plant traits and factors).
The abstract is informative, the manuscript is clearly and concisely written. The method are well described however, I would like the Authors to specify some issues:
− On what basis were the rows selected for the research?
Answer: In the Methods, we describe that the irrigation variants included a given section of the row containing 17 trees, randomly determined at the beginning of the experiment (the experiment also included other variants - sections of the rows and adjacent rows, for which the 13C analysis was not performed).
− How were the rows of trees positioned relative to cardinal directions?
− Were all rows of tested trees parallel to each other?
Answer: Thank you for the notice. Information about the direction of the rows (almost exactly south-north) was added to Methods. The rows were parallel; the row distances were the same. Site coordinates are shown in Methods to allow viewing of the set on aerial photographs.
− In which direction of the world were the branches from which the leaf samples were taken?
− Were samples taken from branches facing the same direction?
Answer: We have described the location of the collected samples in more detail.
− The range of the root zone of apple trees is up to 80 cm down. Why the soil moisture changes were not monitored deeper than 30 cm?
Answer: The 13C and root studies were performed took place during the same years, in the orchard, so we did not know the exactly root depth but penetration under 30 cm was expected. Roots (length density) concentrated in the topsoil and only little roots penetrated to deep subsoil (as shown by a low root density). It was expected, the frequent application of relatively small doses of water by drip irrigation will affect mostly the upper soil layers where the dynamics of moisture changes will be the greatest. We agree the monitoring of soil moisture in deep zones would provide useful information, for example about the possible percolation of irrigation/rain water deeper or even out of the root zone.
The range of references is appropriate and properly cited. In overall, after some minor corrections done by the Authors I recommend the manuscript for being published in the journal Agronomy

Reviewer 2 Report
There are relatively many studies on using carbon isotope discriminant value to indicate the water use efficiency of fruit trees. The paper should highlight the innovation point.
Above all, the paper should first provide the data of yield and water use efficiency, then discuss the relationship between the three parameters. At the same time, as an irrigation research article, it should also give relevant quantitative indicators such as yield.
1. Line 52-53, What was the meaning of this sentence? In my opinion, there was a large amount of literature that does comprehensive assessment, meanwhile, this paper did not provide a c.a.
2. Line 61, 13C value is proportional to WUE and does not mean water deficit.
3. Line 85-86, Leaves located on different branches did not belong to the category of leaf age.
4. The carbon isotopes of leaves at different branches were different, which was related to the accumulation of carbon in source and sink. However, what was the relationship between them and the main variable of the paper, namely the amount of irrigation?
5. Analytical leaf stoichiometry was necessary, but what was the significance of analyzing 15N?
6. The graphs and tables of the paper were poor and should be revised.
7. Leaf sampling is very important. The authors need to describe whether the collected leaves include stems or major veins.
Author Response
Dear reviewer,
we appreciate the time and effort that you dedicated to providing feedback on our manuscript and are grateful for the insightful comments on and valuable improvements to our paper. Thanks for the thorough manuscript review.
Sincerely
Jan Haberle
There are relatively many studies on using carbon isotope discriminant value to indicate the water use efficiency of fruit trees. The paper should highlight the innovation point.
Answer: Based on the comment, we added the latest studies. We are fully aware that WUE (plus the quality of production) is a key indicator of irrigation efficiency. In the study, we preferred to cite studies on apples, because the very differences caused by site conditions, variety, irrigation, tree age, different
tree-cutting technology, etc. limit the comparison of results to a certain extent, probably more than in annual crops. Further, usually reviewers are critical when too much space is devoted to topics that were not the object of study (WUE). For this reason, we excluded some citations from the previous version of the manuscript.
Above all, the paper should first provide the data of yield and water use efficiency, then discuss the relationship between the three parameters. At the same time, as an irrigation research article, it should also give relevant quantitative indicators such as yield.
Answer: We have added information on fruit yields. It should be mentioned that the aim of the study was the effect of irrigation rate on 13C leaf discrimination. We fully agree that the relationship between sink (fruit) size and 13C discrimination deserves research attention; it includes fruit number regulation
during growth (thinning), which is regularly carried out in orchards and was also carried out in an experimental orchard. This complex issue is beyond the scope of this study.
Line 52-53, What was the meaning of this sentence? In my opinion, there was a large amount of literature that does comprehensive assessment, meanwhile, this paper did not provide a c.a.
Answer: Thank you for the comment; the sentence was not well formulated, it was rephrased.
Line 61, 13C value is proportional to WUE and does not mean water deficit.
Answer: The sentence is probably too general but the enrichment of 13C as the indicator of water deficit was shown by many authors and some of them used the statement.
The carbon isotopes of leaves at different branches were different, which was related to the accumulation of carbon in source and sink. However, what was the relationship between them and the main variable of the paper, namely the amount of irrigation?
Answer: The effect of irrigation on 13C leaf discrimination was significant but weak (p=0.022), on the average the ET100 was significantly different from ET0 and ET50. The interaction of treatment and leaf position was not significant (p=0.41) so we did not give it much space, if it was your question? It would be
interesting to study in more detail the interaction between sink/fruit load at the level of one particular branch and the effect of water availability on 13C.
Analytical leaf stoichiometry was necessary, but what was the significance of analyzing 15N?
Answer: In crops, the relationships between 13C and 15N discrimination have sometimes been studied and we hoped that the data could contribute/help interpret the effect of irrigation or annual conditions on 13C discrimination. We must agree that these expectations were not fulfilled, but the observed relationships indicate that this issue deserves separate, detailed research, and that is why we included it in our study.
The graphs and tables of the paper were poor and should be revised.
Answer: We tried to improve the appearance of graphs, although we didn't know exactly what needed to be modified, from the comments.
Leaf sampling is very important. The authors need to describe whether the collected leaves include stems or major veins.
Answer: Thank you for the comment, we specified the leaf sampling in the Methods. We used the term “petiole” rather than stem according to literature (“In botany, the petiole is the stalk that attaches the leaf blade to the stem”

Round 2
Reviewer 2 Report
1. In the current version, it is not easy to identify what the author has changed. Please mark the changed content in red.
2. The drawing of the paper is rough and cannot meet the publication requirements. It is suggested to redraw them.
Author Response
Dear reviewer,
we thank you for your time, effort and valuable comments that allowed us to improve the quality of our manuscript.
- In the current version, it is not easy to identify what the author has changed. Please mark the changed content in red.
In the editing system, we inserted the corrected manuscript in pdf and also in Word in revisions, where all corrections are visible. We will ask the editor if you have access to this file or if you only received the final version with accepted changes.
2. The drawing of the paper is rough and cannot meet the publication requirements. It is suggested to redraw them.
Figure 4 was redrawn and saved in tiff format, with a resolution of 350 dppi as required by the editors.
Round 3
Reviewer 2 Report
The article has been revised and agreed to accept.